# RNAi-Mediated Knockdown of Cottontail Rabbit Papillomavirus Oncogenes Using Low-Toxicity Lipopolyplexes as a Paradigm to Treat Papillomavirus-Associated Cancers

**DOI:** 10.3390/pharmaceutics15102379

**Published:** 2023-09-25

**Authors:** Uzma Ali, Michael Bette, Ghazala Ambreen, Shashank R. Pinnapireddy, Imran Tariq, André Marquardt, Boris A. Stuck, Udo Bakowsky, Robert Mandic

**Affiliations:** 1Department of Pharmaceutics and Biopharmaceutics, Philipps-Universität Marburg, 35037 Marburg, Germanyimran.pharmacy@pu.edu.pk (I.T.); 2Department of Otorhinolaryngology, Head and Neck Surgery, University Hospital Marburg, Philipps-Universität Marburg, 35043 Marburg, Germany; 3Institute of Anatomy and Cell Biology, Philipps-Universität Marburg, 35037 Marburg, Germany; 4CSL Behring Innovation GmbH, 35041 Marburg, Germany; 5Punjab University College of Pharmacy, University of the Punjab, Lahore 54590, Pakistan; 6Department of Pathology, Klinikum Stuttgart, 70174 Stuttgart, Germany

**Keywords:** head and neck cancer, HNSCC, VX2 carcinoma, papillomavirus, CRPV, E6, E7, lipopolyplexes, RNAi

## Abstract

The cottontail rabbit papillomavirus (CRPV)-associated VX2 carcinoma of the New Zealand White rabbit serves as a model system for human papillomavirus (HPV)-associated head and neck squamous cell carcinomas (HNSCCs). The aim of this study was to evaluate the tumor-inhibiting effect of RNAi-mediated knockdown of the CRPV oncogenes, E6 and E7, using siRNA-loaded lipopolyplexes (LPPs). VX2-carcinoma-derived cells were cultured for up to 150 passages. In addition, CRPV E6 and E7 oncogenes were transiently expressed in COS-7 cells. Efficiency and safety of LPPs were evaluated in both VX2 cells and the COS-7 cell line. Both of these in vitro CRPV systems were validated and characterized by fluorescence microscopy, Western blot, and RT-qPCR. Efficient knockdown of CRPV E6 and E7 was achieved in VX2 cells and COS-7 cells pretransfected with CRPV E6 and E7 expression vectors. Knockdown of CRPV oncogenes in VX2 cells resulted in reduced viability, migration, and proliferation and led to a G0/G1 block in the cell cycle. CRPV E6 and E7 siRNA-loaded LPPs could represent promising therapeutic agents serving as a paradigm for the treatment of papillomavirus-positive cancers and could be of value for the treatment of CRPV-associated diseases in the rabbit such as papillomas and cancers of the skin.

## 1. Introduction

Head and neck squamous cell carcinoma (HNSCC) represents the sixth most common cancer, being responsible for 1–2% of all cancer deaths [1,2]. Approximately 50% of HNSCC patients have a 5-year life expectancy of 10–40% [3]. High-risk human papillomavirus (HPV) has been implicated in the pathogenesis of HNSCCs, particularly oropharyngeal cancer. For more than two decades, the auricular VX2 carcinoma of the New Zealand White (NZW) rabbit serves as an animal model for HNSCC [4,5] and is deployed in the evaluation of novel anticancer therapies as well as diagnostic procedures [6]. Advantages of this model are the ease of transplantation, rapid tumor growth, and the presence of local and distant metastatic spread. VX2 tumor development is associated with the cottontail rabbit papillomavirus (CRPV) and was first characterized in 1933 by Shope and Hurst [7]; therefore, it is also known as the Shope papillomavirus. CRPV-induced rabbit papillomas are comparable in etiology and mechanism to many naturally occurring lesions induced by HPV [8,9]. CRPV, like HPV, induces papillomatosis, possibly leading to the development of squamous cell carcinomas. The metastatic pattern of VX2 tumors mimics the natural pattern seen in HPV-associated human HNSCC [10]. In this context, the VX2 model helps us to better understand HPV-associated tumors. VX2 carcinoma cells, therefore, can be considered as an equivalent to HPV-positive HNSCC cells. From this perspective, it is of paramount importance to establish a papillomavirus-positive VX2 cell line which would allow us to perform in vitro studies, thereby helping to reduce animal experiments according to the 3R principle [11]. Several groups reported the generation of VX2-carcinoma-derived cell lines such as VX7, VX-T, and VX-R. However, these cell lines exhibited fewer anaplastic characteristics, loss of transplantability in the host, or loss of ability to establish VX2 tumors [12,13,14,15,16]. At the moment, VX2 carcinoma is available as frozen tissue or a serially transplantable VX2 tumor suspension [17]. Notably, HPV-positive cell lines are underrepresented in numbers, likely due to their lower survival rate in culture as compared to HPV-negative cell lines, which limits studies aiming to understand the biology of papillomavirus-associated tumors [18]. Since the VX2 carcinoma rabbit tumor serves as a model system for HNSCC, we were interested in establishing a VX2-tumor-derived cell line as well as a CRPV E6/E7 transiently expressing cell model to provide a platform for the design and development of antiviral therapies and to comply with the 3R rule. For the latter one, we deployed the widely used monkey-kidney-derived cell line COS-7 [19]. CRPV encodes two types of E6 proteins, a short (SE6) and a long (LE6) version which is equivalent to the E6 protein expressed by HPV-16 and HPV-18. It was observed that CRPV E6 (SE6 and LE6) does not interact with the E3 ligase E6AP and p53 and therefore cannot initiate p53 degradation. Recently, it has been shown that CRPV and HPV-38 E6 interact with p300 (a histone acetyltransferase), which inhibits p53-mediated apoptosis [20]. Studies reported that the CRPV E7 protein interferes with different functions of the retinoblastoma protein, as seen for HPV-16 and HPV-18 [21]. Similarly to that observed for HPV-associated HNSCC tumors, CRPV-related oncogenes E6 and E7 are implicated in viral replication, transformation, tumor growth, and progression, and thus represent potential targets for therapeutic approaches [22]. Thus, in this study, particular emphasis is laid on CRPV E6 and E7 as therapeutic targets. Therefore, siRNA-mediated knockdown of these oncogenes is evaluated as a treatment option for CRPV-positive carcinomas. To achieve successful and efficient gene knockdown, the most crucial step is the selection of appropriate, most effective, and safe gene delivery systems. In recent years, much attention has been given to nonviral delivery systems since they elicit fewer immune responses while enabling unrestricted packaging capacity and ease of synthesis as well as cost-effectiveness and improved targeting potential [23]. Lipopolyplexes, a second-generation nonviral gene delivery system with a size range of 100–200 nm, possess high transfection efficiencies, stability, and improved biocompatibility [24]. Lipopolyplex formulations (DOPE:DPPC:cholesterol) possess higher transfection efficiencies in various cancer cell lines [25]. PEI (polyethylenimine)-based lipopolyplexes [26] were used as transfecting reagents for transient transfections. One main aim of this study was to elucidate the cellular uptake and cytotoxicity level of lipopolyplexes in VX2 and COS-7 cells. Transiently transfected cells were then used to study various biological processes associated with short-term gene expression or gene inhibition (RNAi-mediated gene silencing). This study therefore evaluates siRNA-loaded lipopolyplexes as a treatment option for papillomavirus-associated carcinomas.

## 2. Materials and Methods

### 2.1. Cells and Cell Culture

The African Green Monkey-derived SV40-transformed kidney fibroblast cell line COS-7 was cultivated in Dulbecco’s Modified Eagle Medium (DMEM, PAA Laboratories, Pasching, Austria) supplemented with 10% fetal calf serum (FCS, Sigma Aldrich Chemie GmbH, Taufkirchen, Germany), 2 mmol/L l-Glutamine, 100 U/mL penicillin/streptomycin (both Capricorn, Ebsdorfergrund, Germany), 50 μg/mL gentamicin, and 50 μg/mL amphotericin (both Biochrom, Berlin, Germany) at 37 °C, 5% CO_2_, in a humidified atmosphere. VX2 cells were derived from a solid VX2 tumor that was excised from a tumor-bearing rabbit as described previously [18] and grown in DMEM/Ham’s F-12 medium (Capricorn) containing 10% FCS (Sigma Aldrich Chemie GmbH) 100 U/mL penicillin/streptomycin (Capricorn), 50 μg/mL gentamicin, and 50 μg/mL amphotericin (both Biochrom) in a humidified atmosphere at 37 °C, 5% CO_2_. Surviving cells appeared to be mostly adherent and were considered to represent mainly VX2 tumor cells since normal cells should have become senescent. Both cell lines were grown in 100 mm tissue culture dishes and passaged after reaching 80–90% confluency. To differentiate epithelial VX2-derived tumor cells from fibroblasts, cells from early (4th to 8th) and late (40th to 50th) passages were investigated by immunocytochemistry using an antibody directed against the mesenchymal cell marker vimentin. The nucleus was counterstained with DAPI (1 µg/mL in PBS; Roche Diagnostics, Indianapolis, IN, USA) for 10 min. The signals were documented by confocal laser scanning microscopy (Leica TCS SP2, Leica Microsystems AG, Wetzlar, Germany).

### 2.2. Preparation of Lipopolyplexes

Liposomes, polyplexes, and lipopolyplexes (LPPs) were prepared by a previously established method [17]. Briefly, lipids including DOPE (1,2-Dioleoyl-sn-glycero-3-phosphoethanolamine), DPPC (1,2-Dipalmitoyl-sn-glycero-3-phosphocholine) (both Lipoid GmbH, Ludwigshafen, Germany), and cholesterol (Sigma Aldrich Chemie GmbH, Taufkirchen, Germany), in a proportion of 70:15:15, respectively, were dissolved in 2 mL of a 2:1 (*v/v*) chloroform: methanol solution in a 5 mL round bottom flask. Once dissolved, they were evaporated at 40 °C on a rotary evaporator (Laborota 4000, Heidolph Instruments, Schwabach, Germany) equipped with a vacuum pump, to obtain a thin film. Then, 10 mmol/L HEPES buffer (pH 7.4) was used to hydrate the lipid film by sonication in a bath sonicator to obtain a homogeneous suspension of liposomes. For further size reduction, these liposomes were extruded 21 times from an Avanti Mini Extruder by using polycarbonate membranes (Whatman, Maidstone, UK) of 400 nm and 200 nm pore size, respectively, followed by filtration through 0.2 μm syringe filters. Polyplexes were prepared with a N/P ratio (ratio of polyethylenimine (PEI) nitrogen atoms to nucleic acid phosphate atoms) of 9.5. PEI was diluted in 10 mmol/L HEPES buffer and transferred into 1.5 mL reaction tubes containing an equal volume of DNA or siRNA (diluted in 1× siRNA dilution buffer) followed by incubation at RT for 20–25 min. For the preparation of LPPs, a liposome-to-PEI mass ratio (0.39:1) was used, and calculated amounts of liposomes and polyplexes were vigorously mixed. Complexes were incubated at RT for 1 h and resulting LPPs were used for transfection by adding them to the respective cell culture medium.

### 2.3. RT-qPCR Detection of CRPV E6 and E7 Transcripts

Total RNA from VX2 tumor tissue (positive control) was extracted with the RNeasy FFPE kit (Qiagen, Hilden, Germany), whereas total RNA from rabbit keratinocytes (negative control) and VX2 or COS-7 cells was extracted using the RNeasy Mini kit (Qiagen, Germany) according to the manufacturer’s protocol. RNA concentration was measured with the NanoDrop ND-1000 system (peqLab Biotechnologie GmbH, Erlangen, Germany). cDNA was prepared by using 1.0 µg of total RNA for each sample deploying the Transcriptor First Strand cDNA synthesis kit (Roche, Mannheim, Germany) according to the manufacturer’s protocol. Real-time quantitative PCR analysis (QuantStudio^TM^ 5 system, Thermo Fisher Scientific, Waltham, MA, USA) was performed using the PowerUp^TM^ SYBR Green Master Mix (Applied Biosystems, Darmstadt, Germany) according to the manufacturer’s protocol. The primer length was set between 17 and 27 nt with an optimum length of 20 nt [19]. *GAPDH*, *RPL32,* and *RPLPO* of the rabbit were used as housekeeping genes to normalize gene expression levels of CRPV E6 and CRPV E7 (Appendix A).

### 2.4. Western Blot Analysis

SDS PAGE and Western blot analyses were performed under standard conditions. Whole-cell protein lysates were created through exposing cells to a lysis buffer (20 mmol/L Tris/HCl pH 7.5; 137 mmol/L NaCl; 10% glycerol; 1% NP40; 2 mmol/L EDTA) supplemented with 100 μL/mL Protease Inhibitor Cocktail for Mammalian Cell Extracts (cat# P8340; Sigma-Aldrich Inc., Saint Louis, MO, USA) and 50 μL/mL Phosphatase Inhibitor Cocktail 2 (cat# P5726; Sigma-Aldrich Inc.). Lysis was carried out for 60 min at 4 °C, and the supernatant was harvested after spinning the lysate (>12,000× *g*) for 10 min at 4 °C. Protein concentration was measured with the Bradford method (Bio-Rad Protein Assay Dye Reagent Concentrate; cat# 5000006; Bio-Rad Laboratories GmbH, Feldkirchen, Germany), and 20 μg of whole-cell protein lysate was separated in a 12% SDS polyacrylamide gel followed by transfer onto a nitrocellulose membrane. Precision Plus Protein™ Standards (Bio-Rad, Hercules, CA, USA) were used as a size control. Subsequently, to detect the protein of interest, membranes were blocked for 20 min at RT in 3% skim milk/PBS (Merck, Darmstadt, Germany) followed by overnight incubation with the respective primary antibody at 4 °C. After overnight incubation, the membranes were washed thrice (10 min each) in blocking buffer and incubated with an appropriate HRP-coupled secondary antibody for 1 h at RT. After repeated washing, specific bands were visualized on X-ray film (Agfa, Cologne, Germany) using the enhanced chemiluminescence (ECL) method (Amersham Biosciences, Buckinghamshire, UK). Used antibodies were mouse monoclonal anti-β-Actin (clone AC-74, Sigma-Aldrich, Inc., Saint Louis, MO, USA), mouse monoclonal anti-GFP (clone B-2, sc-9996); rabbit polyclonal anti-RFP (Living colors DsRed Polyclonal antibody, Takara, Kusatsu, Japan), mouse monoclonal anti-PCNA (clone PC10: sc-56), and mouse monoclonal anti-vimentin (clone V9, DAKO, Santa Clara, CA, USA). Mouse IgGκ light chain binding protein (m-IgGκ BP) conjugated to horseradish peroxidase (HRP) (sc-516102, 1:2000) and goat anti rabbit-IgGk HRP (sc-2004, 1:2000), all from Santa Cruz Biotechnology (Santa Cruz, CA, USA), were used as secondary antibodies.

### 2.5. Generation of CRPV E6 and E7 Expression Plasmids

The primers, summarized in Appendix A, were designed to amplify the complete coding sequence of CRPV E6 and E7 using mRNA derived from the VX2 tumor [20]. For this, total RNA from VX2 tumor tissue was isolated with the RNeasy Mini kit (Qiagen, Germany) followed by reverse transcription using the Transcriptor First Strand cDNA synthesis kit (Roche, Mannheim, Germany). PCR was performed using the REDTaq^®^ ReadyMix^TM^ PCR Reaction Mix (Sigma-Aldrich, St. Louis, MO, USA) to generate full-length E6 and E7 cDNA for cloning into the pcDNA3.0, pEGFP-C1, and pDsRed-monomer-C1 vectors (Invitrogen Corporation, Carlsbad, CA, USA), respectively (Appendix A). Double restriction enzyme digestion of the purified E6 and E7 cDNA amplicons and vectors using HindIII/EcoRI (E6, pcDNA3.0), BamHI/EcoRI (E7, pcDNA3.0), and XhoI/EcoRI (E6 and E7, pEGFP-C1, and pDsRed-monomer-C1) was carried out followed by gel purification and ligation overnight at 16 °C using T4 DNA ligase (New England Biolabs GmbH, Frankfurt, Germany). Positive plasmids containing the respective viral cDNA (Appendix A) were further validated by DNA sequencing (4base lab GmbH, Reutlingen, Germany) and subsequent sequence analysis (BioEdit software, v 7.0.5.3; Clustal Omega, https://www.ebi.ac.uk/Tools/msa/clustalo/) (Supplementary Appendix A). Please note that pcDNA3 E6/E7 constructs were generated but not deployed in the present study. Here, it is noteworthy to point out that CRPV E7 harbors an internal HindIII site; therefore, BamHI was used for cloning instead. For plasmid transfection, cells were grown on 6-well plates (1 × 10^6^ cells per well) for 24 h until reaching 60% confluency. Prior to transfection, the cell medium was exchanged with 900 µL of serum-free DMEM/Ham’s F-12 medium (Capricorn) for VX2 cells and serum-free DMEM medium (PAA Laboratories GmbH, Cölbe, Germany) in the case of COS-7 cells, respectively. Each well received 2 μg of plasmid/lipopolyplexes and 800 µL of fresh media. Cells were left incubating for 4 h before adding 1 mL complete media (supplemented with serum and antibiotics).

### 2.6. Expression of CRPV E6 and E7 in VX2 and COS-7 Cells

VX2 and COS-7 cells were grown on coverslips in 6-well plates and transfected with GFP_E6-, GFP_E7-, RFP_E6-, or RFP_E7-expressing plasmids. After incubation for 48 h, cells were washed thrice with PBS (pH 7.4, containing Ca^2+^ and Mg^2+^) and fixed in ice-cold (−20 °C) methanol for 5 min. Then, coverslips were incubated for 20 min with 0.1 µg/mL DAPI (Roche Diagnostics, Indianapolis, IN, USA) to counterstain the cell nucleus. Fluorescence analysis and documentation was performed by confocal laser scanning microscopy (Zeiss Axiovert 100M/LSM 510, Carl Zeiss GmbH, Jena, Germany). To evaluate expression of recombinant proteins, Western blot analysis was performed. Briefly, VX2 and COS-7 cells were seeded on 6-well plates and transfected with green (GFP_E6, GFP_E7) or red fluorescent protein (RFP_E6, and RFP_E7) tagged E6- or E7-expressing plasmids. The original pEGFP-C1 and pDsRed-Monomer-C1 vectors were used as a transfection control (not shown). SDS-PAGE and Western blot analysis were performed as described above.

### 2.7. MTT Cell Viability Assay

To estimate the extent of cellular toxicity following LPP exposure, VX2 and COS-7 cells were seeded in 96-well plates at a density of 2 × 10^3^ and 4 × 10^3^ cells per well, respectively, and transfected with 0.25 µg of plasmid DNA (E7_GFP) per well by either using LPPs or Lipofectamine^TM^ 2000 (LF). Forty-eight hours after transfection, plates were washed twice with PBS (containing Ca^2+^ and Mg^2+^) followed by addition of 200 µL (2 mg/mL MTT; Sigma Aldrich Chemie) reagent per well, followed by incubation for additional 4 h to allow formazan formation. Once formazan crystals were formed, depending on the presence of viable cells and MTT interaction, medium was removed from the wells and replaced with 200 µL of DMSO. Plates were shaken for 30 min at 120 rpm and the absorbance was measured at 570 nm in a FLUOStar^TM^ Optima plate reader (BMG Labtech, Ortenberg, Germany) [21].

### 2.8. RNAi Knockdown of CRPV E6 and E7 by siRNA-Loaded Lipopolyplexes

Three different types of siRNAs were selected from genome sequences of CRPV E6 and E7, as shown in Appendix A. For siRNA transfection, cells were grown on 6-well plates (1 × 10^6^ cells per well) for 24 h until reaching 60% confluency. Prior to transfection, the cell medium was changed to 900 µL of serum-free DMEM/Ham’s F-12 medium. A total of 100 µL of siRNA-loaded LPPs was added to the cells with subsequent incubation for 4 more hours followed by adding 1 mL complete media (with serum and antibiotics). After 48 h incubation, cells were trypsinized and pellets were saved at −80 °C for RT-qPCR and Western blot analysis. Knockdown efficiency was monitored by RT-qPCR. The effect of RNAi on cellular viability was evaluated with the MTT assay. To confirm that siRNAs against CRPV E6 and E7 oncogenes have no major off-target effects and are suitable for specific gene knockdown, a series of experiments was carried out using the VX2-unrelated and CRPV-negative cell line COS-7 that is widely deployed in cell biology studies. COS-7 cells were seeded in 6-well plates at a density of 1 × 10^6^ cells per well in DMEM medium with 10% FCS and antibiotics. After reaching 60–70% confluence, COS-7 cells were transfected with 2 µg of CRPV_E6_GFP- or CRPV_E7_GFP-expressing plasmids, using LPPs as transfecting reagents. After 24 h incubation, fluorescence microscopy (Zeiss Axiovert 100 M, Carl Zeiss Microscopy GmbH, Jena, Germany) was used to evaluate transfection efficiency. COS-7 cells were then transfected with CRPV_E6 or CRPV_E7 siRNA using siRNA-loaded LPPs as described above. Nontarget siRNA was used as a control to account for unspecific “off-target” effects. The expression levels of CRPV_E6 and CRPV_E7 in transiently transfected COS-7 cells were notably higher than in VX2 cells. Nontarget siRNA was used as a control in the same concentration as that used for E6/E7 specific knockdown. Subsequently, cells were trypsinized and cell pellets were stored for RT-qPCR and Western blot analysis, respectively. For flow cytometry analysis, COS-7 cells were fixed in 70% ethanol. RT-qPCR and Western blot analyses were performed using the same protocols as described above.

### 2.9. Measuring CRPV E6 and E7 Knockdown in COS-7 Cells by Flow Cytometry

RNAi knockdown of CPRV E6 and E7 in COS-7 cells should also cause a decline in GFP fluorescence, which serves as a tag for both viral oncoproteins. To evaluate this, GFP fluorescence in COS-7 cells, transfected with CRPV_E6 and CRPV_E7 siRNA-loaded LPPs, was measured by flow cytometry. For this, COS-7 cells were fixed in 70% ethanol and centrifuged for 10 min at 300 xg. After removing the ethanol, the pellet was washed twice in PBS and resuspended in 500 µL PBS followed by flow cytometry analysis (BD LSR II, Becton Dickinson, Franklin Lakes, New Jersey). Dead cells were excluded from analysis using forward- and side-scatter parameters. The GFP signal was detected at 530 nm (30 nm bandwidth filter). Analysis was performed with the FlowJo™ software (version 7.6.5, Tree Star Inc., Ashland, OR, USA).

### 2.10. Wound Healing Assay

VX2 cells were seeded at a density of 1 × 10^6^ cells per well in a 6-well plate in DMEM/Ham’s F-12 medium containing serum and antibiotics. After 24 h, the medium was replaced with 900 µL of serum-free DMEM/Ham’s F-12 medium followed by the addition of 100 µL siRNA-loaded LPPs into each well (nontarget siRNA 50 nmol/L, E6 siRNA 50 nmol/L, E7 siRNA 50 nmol/L, and E6 + E7 siRNA 25 + 25 nmol/L). After 4 h, 1 mL DMEM/Ham’s F-12 medium was added. Twenty-four hours post siRNA transfection, the layer of cells was scratched by using a 100 µL sterile pipette tip. Cells were washed with PBS to remove dislodged cells. After adding fresh medium, cells were left in the incubator for 24 more hours. Subsequently, micrographs were taken at three different positions and analyzed with the Fiji ImageJ software (version 1.53e). The rate of migration was defined as the percentage of wound closure area [27].

### 2.11. Real Time Cellular Analysis (RTCA)

The xCELLigence Real-Time Cell Analyzer^®^ (Roche, Mannheim, Germany) was used to measure growth characteristics such as proliferation rate and cell adhesion of VX2 cells, in which the E6 gene was knocked down using siE6 RNA. The xCELLigence^®^ technology is based on impedance sensing where change in impedance is reported as a dimensionless parameter called cell index (CI; CI = impedance at time point n − impedance without cells/nominal impedance value). The magnitude of the CI depends upon the number of cells, their size, and the degree of firmness of cell adhesion to the substrate coating the plates [28]. Briefly, 96-well E-plates^®^ (Roche), specifically designed for impedance measurement of cells, were prepared by adding 150 μL of DMEM/Ham’s F-12 medium containing serum and antibiotics into each well. Equilibration was achieved by placing the plate into the xCELLigence station, and baseline electrical resistance was gauged to confirm that all connections to the wells were working at appropriate limits. The E-plate was taken out and 50 μL of untreated VX2 cell suspension was added to each well with a cell density of 4.5 × 10^3^ cells per well. Subsequently, an impedance measurement was performed over a period of 24 h at 37 °C. The CI was calculated every 15 min. Once the VX2 cells began to adhere to the support, the cell culture medium was removed and replaced with 175 μL of fresh DMEM/Ham’s F-12 medium and 25 μL of an LPP solution loaded with 150 nmol/L NT or siE6. The CI was measured in real time every 15 min for an additional 72 h.

### 2.12. Flow Cytometry and Cell Cycle Analysis of VX2 Cells

For cell cycle analysis, VX2 cells were transfected with LPPs loaded with siRNA directed against the CRPV E6 and E7 oncogenes. Forty-eight hours after transfection, cells were trypsinized and fixed for at least 2 h in ice-cold (−20 °C) 70% ethanol. After centrifugation (300× *g*, 10 min) and removal of the ethanol, the pellet was resuspended in 500 µL PBS, and RNAse A was added to a final concentration of 50 µg/mL followed by an incubation for 4 h at 37 °C. Propidium iodide (PI; final concentration 5 µg/mL) was added and cells were incubated for 10 min. Cell cycle analysis was performed on a BD LSR II FACS analyzer (Becton Dickinson, Franklin Lakes, NJ, USA) by firstly deploying the forward- (FSC) and side-scatter (SSC) parameters to define a single cell population and to remove cell doublets. Measurement of fluorescence was performed in the PE-H channel. Analysis was performed with the FlowJo™ software (version 7.6.5, Tree Star Inc., Ashland, OR, USA).

### 2.13. Statistical Analysis

All tests were performed using the Graph PadPrism software (version 9.0.0 for MAC; GraphPad Software, Inc., San Diego, CA, USA). The following statistical analyses were used: The one-tailed unpaired Students *t*-test for evaluating the basal expression of E6 and E7 mRNA, the cell viability and the E6/E7 mRNA expression, the FACS analysis of GFP (geometric mean fluorescence) knockdown in COS-7 cells, transiently transfected with GFP-E6- or GFP-E7-expressing plasmids, as well as the cell index measured by real-time cellular analysis; the one-tailed unpaired Mann–Whitney assay for E6/E7 mRNA expression after using different concentrations of siRNA; the two-tailed unpaired T-test with Welch correction for the scratch assay; the two-tailed unpaired T-test for the distribution of cells in each cell cycle phase. Data represent the mean ± SD, with *p* < 0.05 considered statistically significant. Statistical differences were indicated as *: *p* < 0.05, **: *p* < 0.01, ***: *p* < 0.001 and ****: *p* < 0.0001.

## 3. Results and Discussion

### 3.1. Generation of a VX2-Tumor-Derived Cell Line

VX2-tumor-derived tissue was cultured as described above. Early cell culture passages (passage #7) showed a mixture of vimentin (VIM)-positive and -negative cells (Figure 1a). VIM is a typical marker for mesenchymal cells such as fibroblasts. Expectedly, early passages should contain tumor-derived fibroblasts, which after repeated passages should disappear due to senescence. Accordingly, after longer passages (passage #49), no VIM-positive cells could be observed any more (Figure 1a). Earlier, we performed next-generation sequencing (RNASeq) of VX2 tumor tissue-derived RNA [29]. In our present study, we used these data to graphically depict which regions of the CRPV genome are expressed in the VX2 tumor. A major expression is seen for the oncogenes E6 (SE6 and LE6), E7, and E2, with the latter missing part of its coding sequence (Figure 1b). This expression pattern is typically also found in other papillomavirus-associated cancers such as HPV-driven human cancers [30]. Since other papillomavirus-associated genes, e.g., the late genes L1 and L2, are virtually not expressed in these cancers, a major viral assembly and release of complete viral particles from papillomavirus-driven cancer cells appears rather unlikely, although this cannot be ruled out in some cases. To evaluate if the continuously growing, morphologically homogenous, and VIM-negative cells are, indeed, VX2-tumor-derived, we used RT-qPCR to monitor for expression of the major CRPV oncogenes E6 and E7. Both E6 and E7 viral transcripts were found to be significantly expressed in VX2-derived cultured cells, which exhibited a tendency (not statistically significant) to lower expression levels than in VX2 tumor tissues that were carried along as a positive control (Figure 1c). The VX2-derived cell line survived for approximately 150 passages, which allowed for in vitro analysis of these cells in our study. Future studies aim to optimize culturing conditions to obtain an indefinitely growing VX2 cell line. Interestingly, since many more HPV-negative than HPV-positive cell lines exist than would be expected from the frequency of these tumors, it appears likely that establishing HPV-positive HNSCC cell lines is less effective.

### 3.2. Generation of Lipopolyplexes and Evaluation of Cellular Toxicity

Lipopolyplexes (LPPs) (Figure 2a) were generated as described above. The major limitation of most commercially available transfecting reagents is their high cytotoxicity [31]. The MTT assay was deployed to evaluate the effect on cell viability after exposing VX2 and COS-7 cells to LPPs. Cytotoxicity of LPPs was compared to Lipofectamine 2000^TM^ (LF), a well-established in vitro transfection reagent. VX2 cells treated with LPPs exhibited a tendency to higher viability, i.e., lower toxicity, compared to a treatment with LF. This difference reached significance in COS-7 cells (Figure 2b,c). Overall, this points to a lower toxicity of LPPs compared to LPs. These LPPs, therefore, could be promising candidates for a deployment in vivo.

### 3.3. Transfection Efficiency of Lipopolyplexes in VX2 and COS-7 Cells

In order to quantify the transfection efficiency of LPPs, VX2 and COS-7 cells were transfected with CRPV-E7-expressing plasmids. Expression of transcripts was evaluated by RT-qPCR, as shown in Figure 2d,e. Untransfected VX2 and COS-7 cells served as a negative control, whereas cells transfected with LF were used as positive control. CRPV E7 transcripts were detected in highly significant amounts in VX2 as well as COS-7 cells. Expression of E7 transcripts was higher in VX2 cells after transfection with LF, whereas no difference in performance between LPPs and LF was seen in COS-7 cells (Figure 2d,e).

### 3.4. Expression of GFP- and RFP-Tagged CRPV-E6 and -E7 Oncoproteins in VX2 and COS-7 Cells

One drawback when working with CRPV E6 and E7 is the lack of suitable antibodies for detection of these viral oncoproteins. Against this background, we generated chimeric GFP- and RFP-tagged CRPV-E6 and -E7 expression plasmids, which should enable the monitoring of biological effects on E6 and E7 expression in vitro using fluorescence microscopy and flow cytometry. Transient transfection of GFP- and RFP-tagged CRPV-E6- and -E7-expressing plasmids demonstrated pronounced expression levels in COS-7 as well as VX2 cells (Figure 3a,b). Antibodies directed against GFP or RFP could detect GFP-E7 and RFP-E7 chimeric proteins during Western blot analysis in COS-7 and VX2 cells. However, no expression of GFP-E6 and RFP-E6 could be seen in our Western blot settings (Figure 3c). Since GFP-E6 and -E7 were expressed in both COS-7 and VX2 cell lines, it appears likely that the chimeric E6 protein becomes insoluble during sample preparation for Western blot analysis. Interestingly, in a study by Liu et al., the authors described that HPV16-E6 under specific conditions exhibits a tendency to become insoluble [32]. Similar structural features of the CRPV-E6 protein could explain why the GFP/RFP-tagged CRPV-E6 protein does not become soluble during cell lysis for Western blot analysis.

### 3.5. Effectiveness of CRPV E6 and E7 RNAi Knockdown Using siRNA-Loaded LPPs

Selecting suitable siRNA concentrations with significant knockdown potential yet minimum cytotoxicity is a relevant first step. Here, several factors can influence siRNA-mediated gene knockdown, such as choice of the siRNA sequence, concentration, transfecting reagent, and the type of cell line. Next to VX2 cells, the CRPV-negative cell line COS-7 was used as a VX2 cell independent test system for CRPV E6 and E7 RNAi studies (Figure 4). For this, plasmids encoding CRPV_E6 and CRPV_E7 were transiently transfected using low-toxicity LPPs as transfecting reagents. Here, the specificity and mRNA knockdown efficiency of siRNAs directed against CRPV E6 and E7 oncogenes could be assessed in a defined CRPV_E6- and CRPV_E7-expressing cell system (Figure 4a) and compared to VX2 cells (Figure 4b). Efficient CRPV_E6 and CRPV_E7 mRNA knockdown could be observed with 150 nmol/L siRNA in COS-7 cells (Figure 4a). However, VX2 cells exhibited a rather paradoxical behavior, demonstrating a solid CRPV_E6 and CRPV_E7 mRNA knockdown at 50 nmol/L siRNA, whereas higher levels did not reach this efficiency (Figure 4b). This observation could be explained by the fact that there is not always a uniform concentration-dependent cellular uptake of siRNA/lipopolyplexes, which could result in different knockdown efficiencies. Similarly, every N/P ratio of the polymer/siRNA complex can result in a different knockdown efficiency which is not necessarily dependent on the concentration of the siRNA. To evaluate the level of CRPV E6 and E7 protein knockdown using siRNA-loaded LPPs, COS-7 cells expressing E6-GFP or E7-GFP were treated with siRNA directed against E6 (siE6) or E7 (siE7). Flow cytometry analysis could demonstrate a highly efficient knockdown of (GFP-)E6 and (GFP-)E7 measured by the respective reduction in fluorescence intensity, which corresponded to the downmodulation of E6 and E7 mRNA levels as seen during RT-qPCR analysis (Figure 4c). PCNA (proliferating cell nuclear antigen) serves as a reliable proliferation marker that is responsible for regulating the process of DNA replication during the S-phase of the cell cycle, thereby having a vital role in cell proliferation. Western blot analysis was performed to evaluate the effect of CRPV E6 and E7 RNAi knockdown on PCNA expression. Western blot analysis of E6/E7-siRNA-treated cells showed a prominent decline in the PCNA protein levels, which suggested a reduced proliferation. Protein bands were quantified based on optical density/intensity using the ImageJ/Fiji software (version 1.54f) [24].

### 3.6. Wound Closure Assay and Real-Time Cellular Analysis (RTCA)

Previously, it was found that knockdown of E6 and E7 alone or in combination elicits an effect on the rate of cell migration in vitro in the HPV-16-positive cervical cancer cell lines Caski and SiHa [33]. To evaluate the influence of CRPV E6 and E7 on cell migration in VX2 cells, we performed a wound healing assay. The rate of migration was defined as the percentage of wound (scratch) closure area. Analysis revealed that 24 h after applying a defined scratch in a confluent VX2 cell layer, closure of the wounded area was significantly lower after RNAi knockdown of E6/E7 (Figure 5a). The data suggest that downregulation of the CRPV oncogenes E6 and E7, individually or combined, inhibits cell migration of VX2 cells. To test whether this effect could be caused by a decreased growth rate after E6/E7 knockdown, the growth curves of VX2 cells with and without knockdown of E6 were measured in a real-time cellular analysis assay (RTCA) (Figure 5b). This RTCA system (xCELLigence) can be used to monitor live cell viability, proliferation, motility, adhesion, migration, invasion, cell number, and morphology. It provides label-free and real-time surveillance of cell viability by tracking the electrical impedance as a readout [34]. The data show a decreased proliferation rate for the entire duration of the electrical impedance recording (72 h) and a significantly decreased proliferation rate after downregulation of E6, indicating an overall decrease in growth rate.

### 3.7. Indication of G1 Block after CRPV E6 and E7 Knockdown in VX2 Cells

The effect of CRPV oncogene RNAi knockdown on the cell cycle of cultured VX2 cells was evaluated by flow cytometry. Consistent with the literature on long-term VX2 cell cultures [35], in untreated VX2 cells, 50.2% (+/−2.0%) of cells were in the G0/G1 resting phase, 32.2% (+/−1.2%) were in the S replication phase, and 17.2% (+/−0.4%) were in the G2/M growth phase (Figure 6a,b). Elimination of CRPV E6 or E7 alone or in combination resulted in a significant arrest of VX2 cells in the G0/G1 phase of the cell cycle compared to the control (NT), which was accompanied by a reduced number of cells in the S and G2/M phases of the cell cycle, which in some cases reached significance (Figure 6b). The cell cycle analysis data correlate with cell viability, which decreased significantly in all three knockdown groups (siE6, siE7, and siE6 + siE7) (Figure 6c), suggesting that E6 and E7 support tumorigenic growth of VX2 cells by promoting entry into the mitotic phase, which is a well-known feature for these oncoproteins [36]. Similarly, previous studies demonstrated a role of the CRPV E6 protein in protecting tumor cells from apoptosis and maintaining cancer cell population integrity [37]. These reports are in agreement with our observations in VX2 cells. Furthermore, various studies have previously demonstrated that CRPV E6 and E7 oncogenes play a vital role in the transformation and anchorage-independent growth of tumor cells [38]. It can, therefore, be expected that RNAi-mediated downregulation of these genes affects the rate of VX2 cell proliferation.

Previous reports indicated roles of CRPV E6 and E7 as key genes for papilloma formation in rabbits and also addressed the importance of these genes as possible therapeutic targets for CRPV-induced carcinomas [29]. Downregulation or suppression of either of these genes may, therefore, cause a significant decline in papilloma development. Another study reported the outcome of CRPV E1, E2, E6, and E7 vaccination, both individually and in combination, in rabbits previously injected with CRPV viral DNA at specific sites. Papillomas grew in all (100%) nonvaccinated rabbits. Rabbits vaccinated with a single-gene vaccine were only partially protected against papilloma growth on the challenged sites. However, two out of four rabbits vaccinated with an E1, E2, E6, and E7 combination vaccine became completely free of papillomas, while in the other two rabbits only small papillomas developed at the challenged sites, which regressed within three weeks. This outcome showed that CRPV oncoproteins, when targeted in combination, may lead to synergistic therapeutic effects [24]. These findings are in agreement with those of our study, which demonstrate that RNAi-mediated knockdown of CRPV E6 and E7 results in reduced viability, migration, and proliferation by inducing a block in the G0/G1 phase of the cell cycle.

In addition to toxicity, the selectivity of gene transfection systems is decisive for their applicability as therapeutic agents in the patient. The lipopolyplexes we use are a further development of the classic polyplexes using the optimum biocompatibility of the lipoplexes. It has been known for many years that polyplexes have a very good transfection efficiency in vitro (e.g., cell culture), but many problems in the living organism include, among other things, incompatibility with blood, poor trafficking to the target site, and accumulation in the liver or lungs. The lipid envelope present in lipopolyplexes should at least reduce such problems. These lipopolyplexes are available as parenteral formulations and are preferably designed for intravenous administration. Studies in animals observed the intravenous injection route to be effective for tumor targeting [24,39]. Good biocompatibility and sufficiently long circulation times enable accumulation in tumor tissues with an EPR (enhanced permeability and retention) effect. The accumulation in the liver should also be significantly reduced. Linder et al. and Kurosaki et al. were able to demonstrate that lipopolyplexes showed a significantly improved transfection efficiency, with only very low accumulation in the liver [40,41]. Similarly, Ali et al. were able to show in a comparable drug delivery system that there was a reduction in liver accumulation after lipid coating [42].

## 4. Conclusions

Our previous studies [5] demonstrated the suitability of the VX2 carcinoma as a model system for HPV-associated HNSCC, since this tumor, similarly to HPV+ HNSCC, is the consequence of a papillomavirus (cottontail rabbit papillomavirus, CRPV) infection. Against this background, it was of utmost importance to implement a CRPV cell culture model that allows us to perform studies in vitro. We demonstrated that transiently cultured VX2 cells are a suitable cell culture model. Furthermore, an artificial CRPV model system, based on transient transfection of CRPV E6 and E7 oncogenes into the well-established cell line COS-7, allows us to perform controlled investigations regarding the effectiveness of candidate therapeutics such as the E6/E7 siRNA-loaded LPPs. Since both cell systems allow many of the essential investigations to be performed in vitro, they help in reducing animal experiments according to the 3R rule (reduce, replace, refine). Lessons learned from these studies could directly lead to the development of analogous therapeutics for the treatment of HPV-positive head and neck tumors in human patients. In the present study, we demonstrated the potential usefulness of E6/E7-siRNA-loaded LPPs as a treatment option for papillomavirus-associated head and neck cancers. Notably, LPPs directed against CRPV E6 and E7, as presented in our study, could be directly used as possible therapeutic agents for diseases associated with CRPV such as rabbit skin papillomas and cancers. This pharmaceutical LPP formulation is designed for parenteral administration (i.v., i.m., and pulmonary). Evaluating the bioavailability and efficacy in vivo of these formulations will be an interesting task for following studies.

## Figures and Tables

**Figure 1 pharmaceutics-15-02379-f001:**
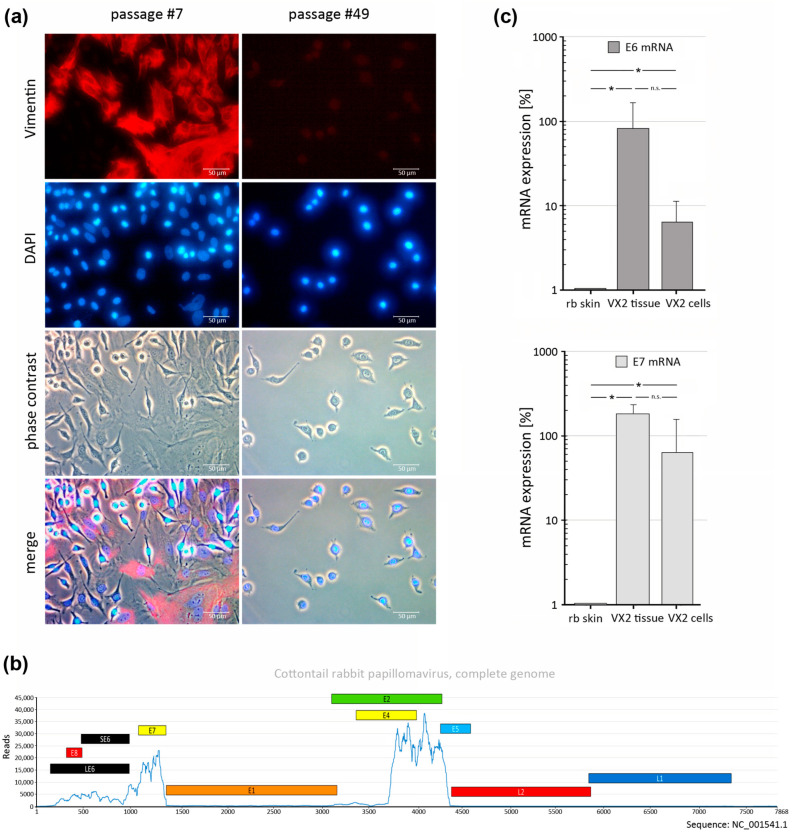
Generation of VX2 cell cultures. (**a**) Immunofluorescence analysis of VX2 cell cultures during early (p #7) and late (p #49) passages demonstrating vimentin-expressing cells at early passages only. (**b**) Pattern of gene expression of the CRPV genome. (**c**) CRPV E6 and E7 mRNA expression levels in VX2-derived cell cultures compared with VX2 tissues. (E1, E2, E4, E5, LE6, SE6, E7, and E8 = “early” genes; L1 and L2 = “late” genes). Statistics: one-tailed unpaired Student’s *t*-test (n = 3 experiments); significance level *: *p* < 0.05, n.s.: no significance.

**Figure 2 pharmaceutics-15-02379-f002:**
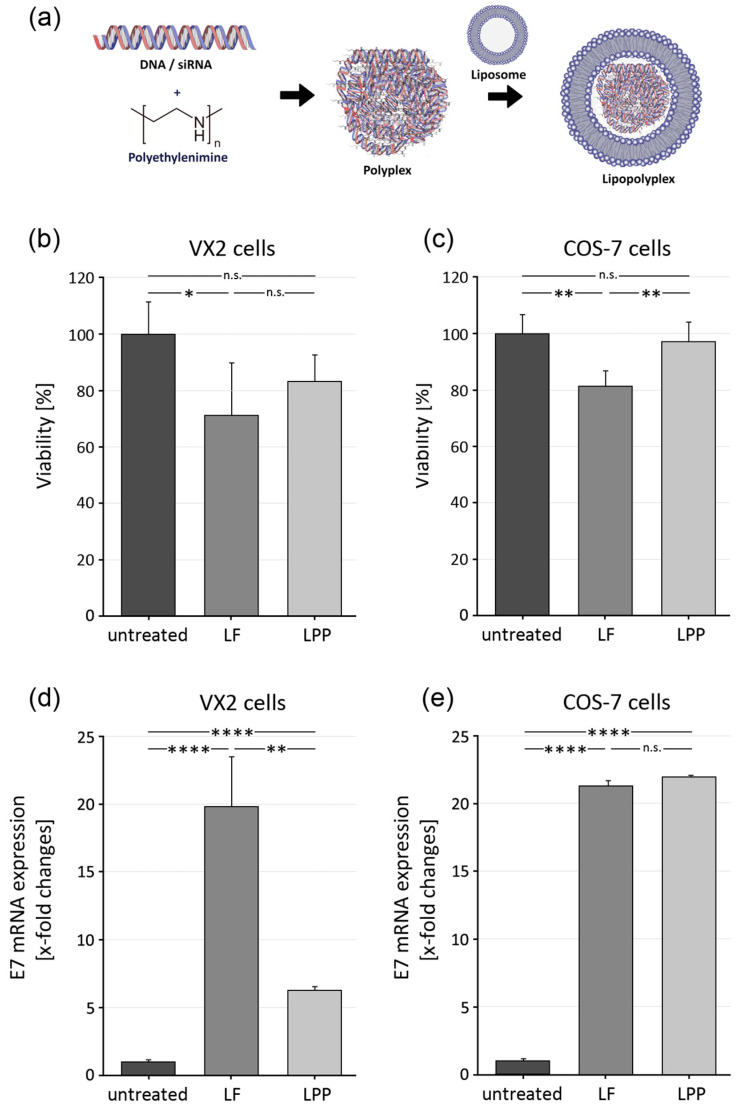
Response of cells to transfection with lipopolyplexes (LPPs) in comparison with Lipofectamine 2000^TM^ (LF). (**a**) Schematic, depicting assembly of LPPs. (**b**,**c**) Effect on viability of (**b**) VX2 and (**c**) COS-7 cells after transfection with LF or LPPs. (**d**,**e**) Level of CRPV E7 mRNA expression after transfecting (**d**) VX2 and (**e**) COS-7 cells with CRPV E7-expressing plasmids using LF or LPPs. Statistics: one-tailed unpaired Student’s *t*-test (n = 3 experiments); significance level *: *p* < 0.05, **: *p* < 0.01, and ****: *p* < 0.0001, n.s.: no significance.

**Figure 3 pharmaceutics-15-02379-f003:**
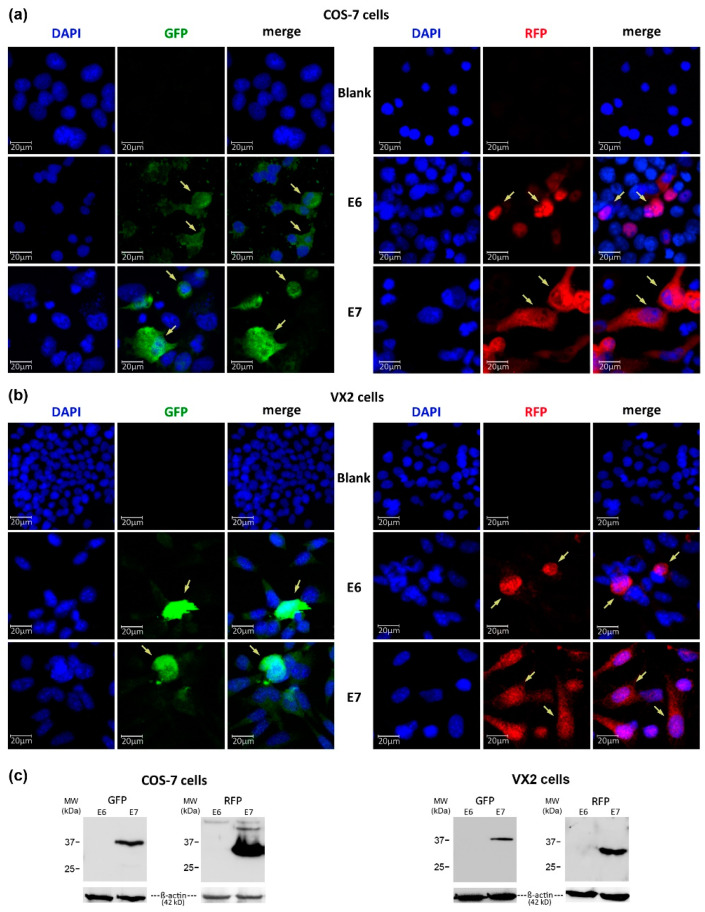
Expression of CRPV oncoproteins E6 and E7 in COS-7 and VX2 cells. Fluorescence microscopic images of COS-7 (**a**) and VX2 (**b**) cells transfected with viral oncoprotein-expressing plasmids tagged with GFP or RFP. GFP_E6 or GFP_E7 expression (arrows) is depicted in the green (GFP) channel, whereas RFP_E6 or RFP_E7 expression (arrows) is seen in the red (RFP) channel. Nontransfected cells (blank) were used for background labeling. (**c**) Western blot analysis, demonstrating expression of GFP-E7 and RFP-E7 recombinant proteins of expected size in transfected COS-7 and VX2 cells. No specific signal is seen for GFP-E6 and RFP-E6.

**Figure 4 pharmaceutics-15-02379-f004:**
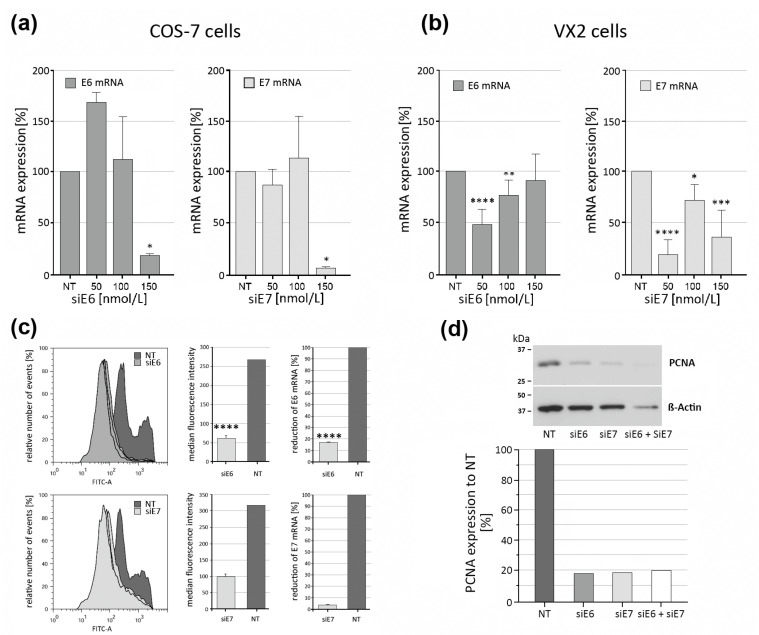
RNAi knockdown of CRPV E6 and E7 using siRNA-loaded LPPs. Effect of CRPV E6 and E7 RNAi knockdown in COS-7 (**a**) and VX2 (**b**) cells transfected with E6- and E7-expressing plasmids, using different siRNA concentrations (50, 100, and 150 nmol/L), on the mRNA expression levels of these oncogenes. (**c**) Prominent downregulation of the GFP signal (geometric mean fluorescence) after E6 and E7 RNAi in COS-7 cells, transiently transfected with GFP-E6- or GFP-E7-expressing plasmids. (**d**) RNAi knockdown of CRPV E6 and E7 results in reduction of PCNA expression. Statistics: (**a**,**b**) one-tailed unpaired Mann–Whitney assay; (**c**) one-tailed unpaired Student’s *t*-test (n = 3 experiments); significance level *: *p* < 0.05, **: *p* < 0.01, ***: *p* < 0.001, and ****: *p* < 0.0001.

**Figure 5 pharmaceutics-15-02379-f005:**
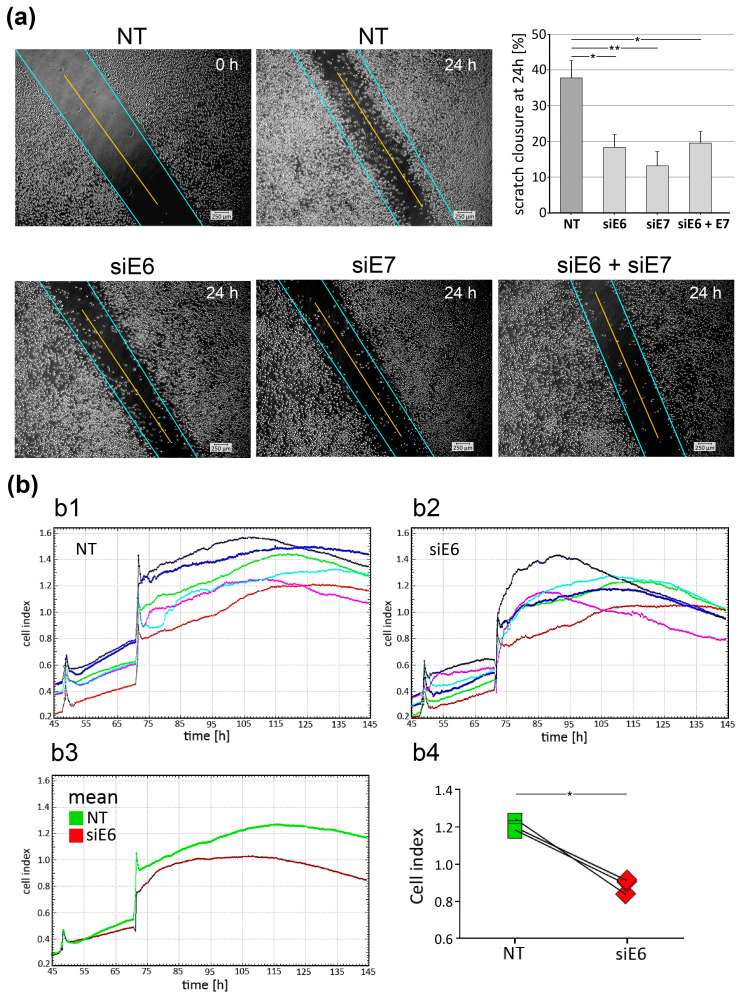
Functional effects of E6/E7 knockdown in VX2 cells on their migratory ability and proliferation rate. VX2 cells were transfected by LPP-carrying nontarget (NT) RNA, siE6 RNA, siE7 RNA, or siE6 + siE7 RNA in combination. (**a**) Representative microscopic images of wound healing assays using VX2 cells are shown. The baseline situation of the wound healing assay at time 0, immediately after the scratch, is shown as an example for the NT siRNA. All other microscopic images show the in vitro cultures 24 h after the scratch. The blue line marks the boundary between open and closed scratch; the yellow line represents the length of the scratch used for the statistical analysis of scratch (wound) closure, as shown in the associated graph. All wound healing experiments were performed in triplicates. (**b**) Real-time cellular analysis of the proliferation rate of VX2 cells after CRPV E6 knockdown. VX2 cells were treated with LPPs, loaded with 50 nmol/L NT control RNA (b1) or siE6 RNA (b2). Each colored line in the top 2 graphs corresponds to a measurement in the xCELLigence Real-Time Cell Analyzer^®^. The lower left graph (b3) shows the mean values from each of the six independent measurements, the lower right graph (b4) depicts the determined proliferation rate. Mean +/− SD are shown in (**a**). Statistical differences of the used two-tailed unpaired *t*-test with Welch correction for (**a**) and the one-tailed unpaired Student’s *t*-test for (**b**) were indicated as *: *p* < 0.05, **: *p* < 0.01.

**Figure 6 pharmaceutics-15-02379-f006:**
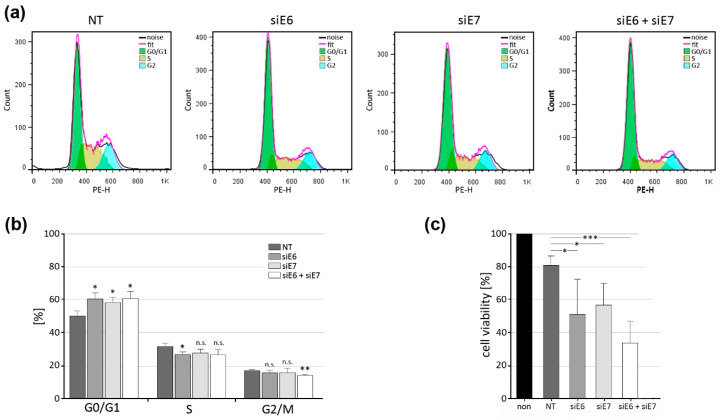
Effect of E6/E7 knockdown on the cell cycle and viability of VX2 cells. (**a**) Representative cell cycle histograms of VX2 cells after RNAi knockdown of the CRPV oncogenes E6 and E7. (**b**) Percentage of VX2 cells in each cell cycle phase according to the respective treatment group. (**c**) Effect on VX2 cell viability after treatment with siE6-, siE7-, or siE6 + siE7-loaded LPPs (“non” indicates the group of untreated VX2 cells; “NT” indicates the group of VX2 cells treated with nontarget-RNA-loaded LPPs). All experiments were performed in triplicate. Means +/− SD are shown. Statistical differences were evaluated with the unpaired two-tailed Student’s *t*-test and (**b**) the unpaired one-tailed Student’s *t*-test (**c**), and are indicated as *: *p* < 0.05, **: *p* < 0.01, and ***: *p* < 0.001, n.s.: no significance.

## Data Availability

The data presented in this study are available on request from the corresponding author.

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
