# Peer review of "RNAi-Mediated Knockdown of Cottontail Rabbit Papillomavirus Oncogenes Using Low-Toxicity Lipopolyplexes as a Paradigm to Treat Papillomavirus-Associated Cancers"

_pharmaceutics, 2023, doi:10.3390/pharmaceutics15102379_

Round 1

Reviewer 1 Report

Dear authors,

head and neck squamous cell carcinoma is a serious cancer and requires a lot of  knowledge for successful treatment.

1. What is ultimate goal fo your research? If in vivo studies bring you good results, what will be your next step? How your end-product will look like, incection, ointment, etc.? Please, explain in conclusion.

2. You used one-tailed unpaired Student's t-test. This method is not strong enough. Could you explain your choice?

3. Figure 2 (b) - has significant difference in comparison with 'LF' because bars of SD are quite big for 'LF'. What were confidence interavals for both groups?

4.  Figure 5. Graphs of 'b' part should be subdivided on b1, b2, b3, and b4, and explained one by one.

5. Figure 6.  - it is not clear that 'NT' has significant difference in comparison with 'siE6' because bars of SD are quite big for 'siE6'. What were confidence interavals for both groups?

6. Figure 4 (a). How you can explain that mRNA expression substantially decreases for COS-7 cells for 150 nmol/L concentration while for VX2 cells (b) this does not occur?

7. Figure 4 (c). For prominent downregulation can you add statistical differences?

8. Figure 4. You indicate asterisks (*). What do they mean, explain in your figure legend?

9.  Figure 2. * - is not explained in the figure legend.

10. Figure 3. Please, add additional arrows to indicate on the graph what reader should pay attention to.

In summation, please, pay special attention to your statistical analysis and figures with figure legends and make all required corrections.

Kind regards

Reviewer 2 Report

This is an interesting study by Ali and colleagues on using lipoplexes to deliver siRNA targeting oncoproteins E6/E7 of the cottontail rabbit papilloma virus as a treatment strategy for HPV-associated head and neck squamous cell cancers.

1). The manuscript is well written with only minor corrections needed (line 62, 437). Methods are explained in detail.

2). The biggest weakness of the manuscript is the knockdown response seen with varying concentrations of siRNAs against E6/E7 in COS-7 and especially VX2 cells (Figure 4-panel a/b). The resulting mRNA levels seen by rt-qPCR in VX2 lines is erratic with 50nM having a larger effect on E6 than 100 and 150nM. More concerning is the almost bimodal response of E7 mRNA reduction with increasing concentrations of siRNA targeting E7. What is the explanation for these results? The authors did not address this at all in the manuscript. 

3). To be clinically relevant, this siRNA targeting E6/E7 need to be delivered to the tumor site. Where would lipoplexes go to if injected into an animal model of HNSCC? My guess would be that the vast majority would be trapped in the liver. This potential drawback needs to be addressed. 

Minor corrections needed. 

Round 2

Reviewer 2 Report

Dear Authors,

Thank you for your response to my major comment #2 and 3. Please add your response to comment #3 to end of your discussion section to give the reader a sense of where the literature on lipoplexes stands, and what you can predict, based on prior studies, about lipoplexes escaping from being trapped in the liver. 

Acceptable
